# Role of epithelial to mesenchymal transition associated genes in mammary gland regeneration and breast tumorigenesis

Shaheen S. Sikandar[1], Angera H. Kuo[1], Tomer Kalisky[2,5], Shang Cai[1], Maider Zabala[1], Robert W. Hsieh[1], Neethan A. Lobo[1], Ferenc A. Scheeren [1,6], Sopheak Sim[1], Dalong Qian[1], Frederick M. Dirbas[3], George Somlo[4], Stephen R. Quake[2] & Michael F. Clarke[1]

Previous studies have proposed that epithelial to mesenchymal transition (EMT) in breast cancer cells regulates metastasis, stem cell properties and chemo-resistance; most studies were based on in vitro culture of cell lines and mouse transgenic cancer models. However, the identity and function of cells expressing EMT-associated genes in normal murine mammary gland homeostasis and human breast cancer still remains under debate. Using in vivo lineage tracing and triple negative breast cancer (TNBC) patient derived xenografts we demonstrate that the repopulating capacity in normal mammary epithelial cells and tumorigenic capacity in TNBC is independent of expression of EMT-associated genes. In breast cancer, while a subset of cells with epithelial and mesenchymal phenotypes have stem cell activity, in many cells that have lost epithelial characteristics with increased expression of mesenchymal genes, have decreased tumor-initiating capacity and plasticity. These findings have implications for the development of effective therapeutic agents targeting tumor-initiating cells.

[1] Institute for Stem Cell Biology and Regenerative Medicine, School of Medicine, 265 Campus Drive, Stanford, CA 94305, USA. [2] Department of Bioengineering, 318 Campus Drive, Stanford, CA 94305, USA. [3] Department of Surgery, Stanford University School of Medicine, Stanford Cancer Institute, 875 Blake Wilbur Drive, Rm CC2235, Stanford, CA 94305, USA. [4] City of Hope Comprehensive Cancer Center, 1500 East Duarte Road, Duarte, CA 91010, USA. [5] Present address: Faculty of Engineering, Bar-Ilan University, Ramat Gan 52900, Israel. [6] Present address: Department of Medical Oncology, Leiden University Medical Center, Leiden RC 2300, The Netherlands. Correspondence and requests for materials should be addressed to M.F.C. (email: mfclarke@stanford.edu)

Epithelial to mesenchymal transition (EMT) is the process by which epithelial cells lose cell–cell contact, detach from basement membranes and acquire more fibroblast-like features. Expression of genes associated with EMT such as Snail1, Snail2, Twist1, Zeb1, Zeb2, and Sox9 in cancer cells have been linked to stem cell properties and metastasis[1–7].

The process of acquiring mesenchymal features or down-regulating epithelial characteristics to acquire 'stem cell' features in normal and tumor cells has been previously suggested. A majority of these studies derived their conclusions mainly from in vitro cell culture models. Although commonly used, these models are associated with the concern that cells have altered behavior due to mutations acquired during the establishment of the cells line, the lack of a three dimensional context, loss of apical basal polarity and lack of in vivo micro-environment signals. Interestingly,

recent studies using mouse transgenic cancer models in combination with S100a4 lineage tracing have suggested that in breast tumors EMT is not responsible for their metastasis to the lung but plays a role in chemo-resistance[8]. Adding further complexity, intra-vital imaging using E-cadherin reporter mice showed that downregulation of epithelial characteristics in a mouse model of breast cancer may enhance migration but does not determine metastatic outgrowth formation[9].

However, the contribution of EMT in acquiring stem cell characteristics is still under debate. It had been initially hypothesized that expression of EMT genes in normal and cancer cells is associated with acquisition of stem cells characteristics[4,10–12], while later reports have suggested a more complex relationship[11,13–16]. However, this hypothesis has not been formally tested in vivo in mouse mammary epithelial cells or patient derived xenografts.

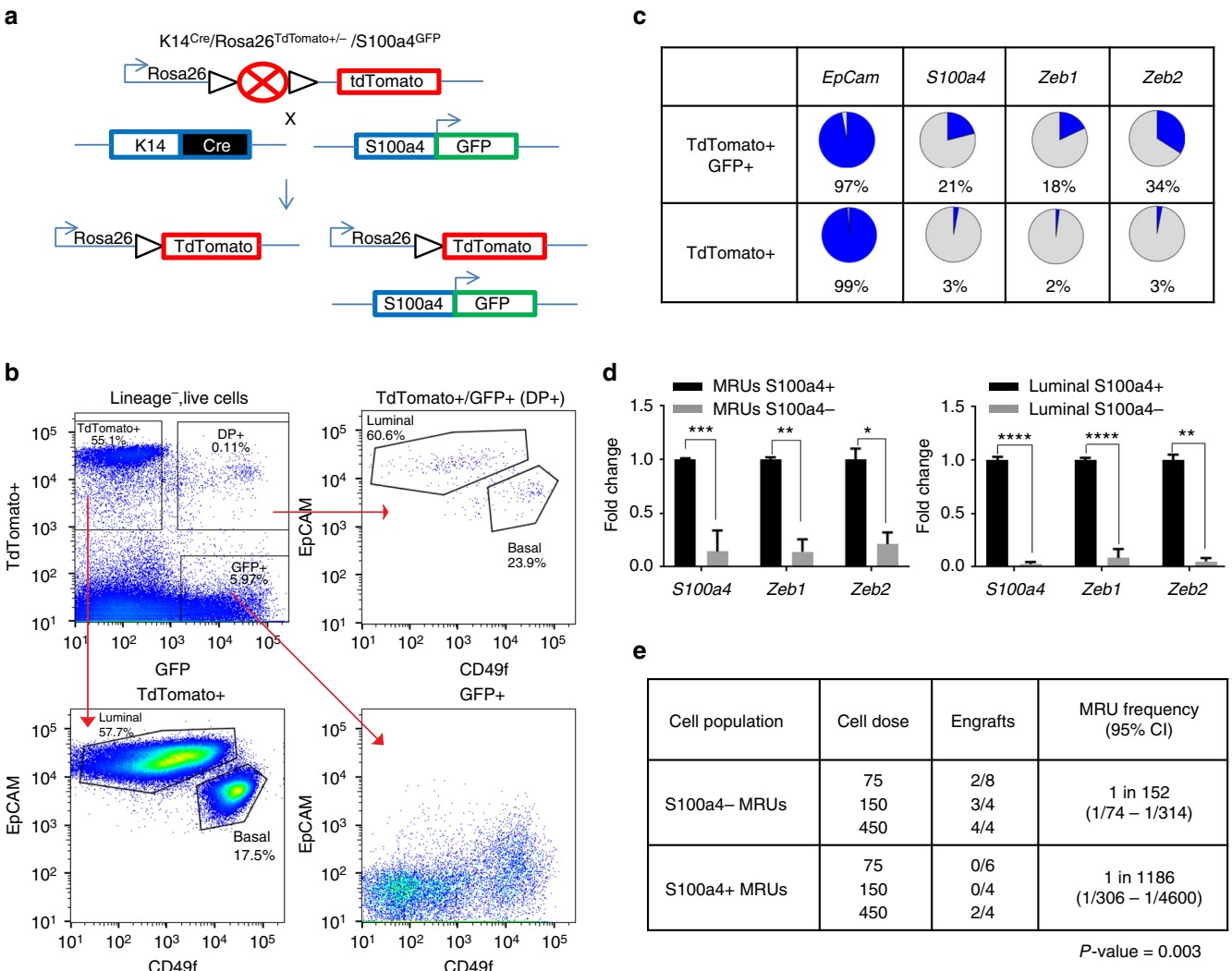

**Fig. 1** Keratin 14+/S100a4+ cells express genes associated with EMT. **a** Schematic diagram showing the generation of the K14^Cre^/Rosa^TdTomato^/S100a4^GFP^ mouse. K14^Cre^ marks all epithelial cells TdTomato+, while S100a4 dynamically controls expression of GFP. **b** FACS analysis of the K14^Cre^/Rosa^TdTomato^/S100a4^GFP^ mouse mammary cells. The cells are gated on lineage^−^ (CD45^−^, CD31^−^, Ter119^−^), DAPI^−^ cells. TdTomato+, GFP+, or TdTomato+/GFP+ double positive (DP) cells are then analyzed using EpCAM and CD49f. TdTomato+ and DP cells are present in both luminal and basal lineages. Note: TdTomato+ cells are at the chart edges to allow for adequate separation from DP cells. (n = 6 mice). **c** Gene expression in single, FACS-isolated TdTomato+/GFP+ and TdTomato+ basal cells were analyzed by qRT-PCR. The results (Supplementary Fig. 1) are summarized here as the percentage of single cells analyzed that express the indicated genes. See Supplementary Fig. 1a for raw data and Supplementary Fig. 1b for expression values of individual genes and p-values. **d** Real time gene expression analysis of S100a4, Zeb1, and Zeb2 in S100a4+ and S100a4− MRUs and luminal cells (n = 3 mice sorted for different populations). The data are shown as mean ± SD. *p < 0.05, **p < 0.01, ***p < 0.001, ****p < 0.0001 (t-test). **e** S100a4+ MRUs and S100a4− MRUs were sorted from 10–12-week-old nulliparous mice and used in transplantation assays in 3-week-old recipient mice. The data are combined from two independent transplant experiments (p = 0.003, frequencies and p-values are calculated using ELDA software)

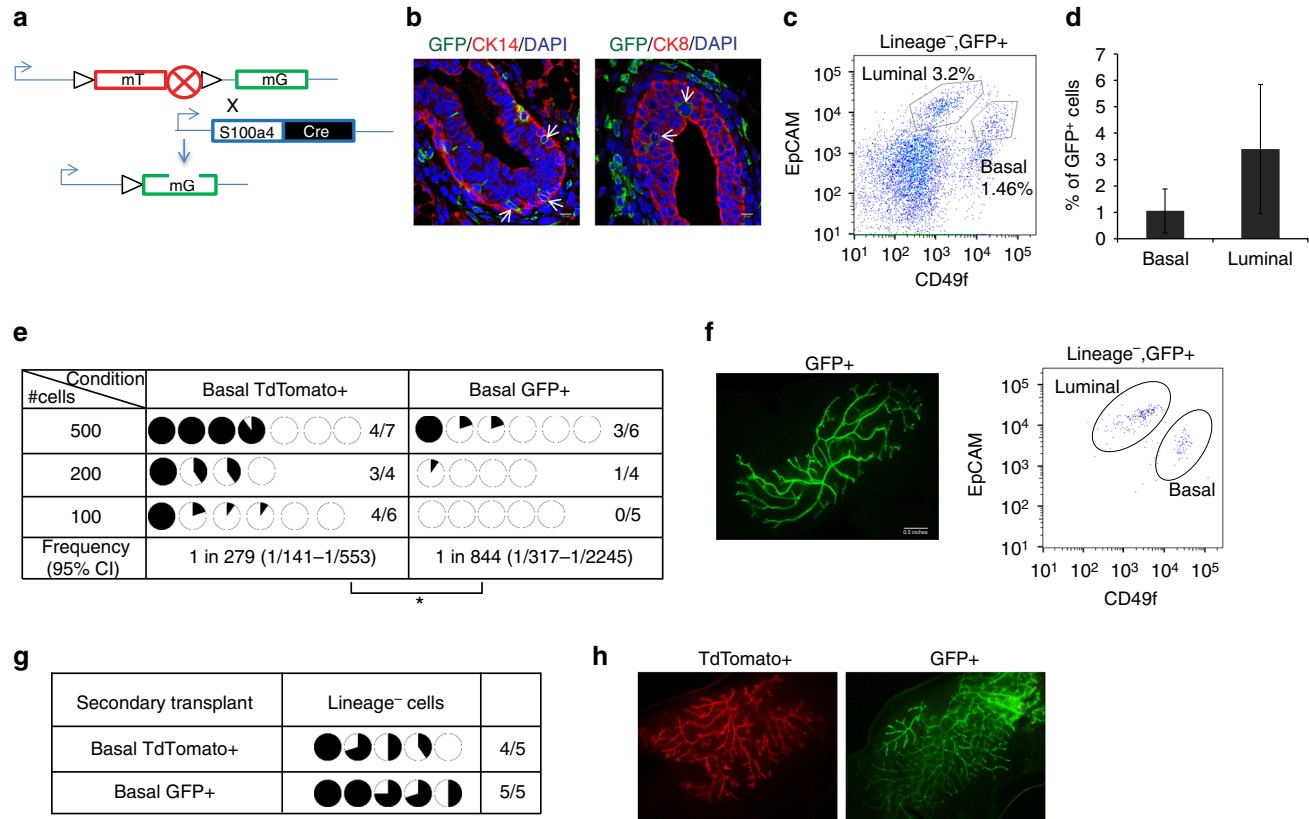

**Fig. 2** S100a4 lineage traced basal cells can regenerate the mammary gland in vivo. **a** Schematic diagram showing the crossing of $Rosa26^{mTmG}$ reporter with $S100a4$-cre mice. **b** Immunofluorescence analysis showing that $S100a4$ lineage traced cells (GFP+) are present in mammary epithelial cells. CK8/CK14 (red), GFP (green), and DAPI (blue). Arrows indicate S100a4 traced epithelial cells. Scale bar, 10 μm. **c** Representative FACS analysis of the $S100a4^{cre}$/$Rosa26^{mTmG}$ mouse mammary gland of Lineage$^-$/Live$^-$/GFP+. ($n = 20$ mice). Also see Supplementary Fig. 2a. **d** Percentage of GFP+ cells in basal and luminal cell populations. The data are shown as mean ± SD ($n = 10$ mice). **e** Basal TdTomato+ and GFP+ cells were sorted from 10–12-week-nulliparous females and transplanted in 3-week old mice. All transplants were analyzed after 8 weeks. The data are combined from three independent transplantation experiments. Pie charts show the percentage of mammary fat pad filled. (*$p < 0.05$, frequencies and $p$-value are calculated using ELDA software). Also see Supplementary Fig. 2 a for sorting scheme. **f** Representative image of a basal GFP+ transplant (left). FACS analysis using EpCAM and CD49f showing basal and luminal cell populations (right) ($n = 3$ outgrowths). **g** Secondary transplant of cells from **e**. Pie charts show percentage of mammary fat pad filled. **h** Representative image of secondary transplants from TdTomato+ (left) and GFP+ basal cells (right) ($n = 4$ outgrowths)

Moreover, deletion/overexpression of genes can sway a delicate system in one direction or the other, leading to conflicting results. We wanted to test if EMT-like cells contribute to normal mammary tissue development and in tumor initiating cells without external perturbations. Specifically, we wanted to test the contribution of EMT-like cells when the rudimentary mammary duct invades the fat-pad to form the mature mammary gland during puberty. Cancer is a caricature of the normal and tumors retain both tissue hierarchy and heterogeneity[17,18]. Previous studies have demonstrated that cancer cells use programs of normal development and self-renewal for tumor formation and maintenance[19]. Hence, we also wanted to test whether tumor cells that express higher levels of EMT-associated genes are enriched in tumor initiating capacity in primary patient samples/patient derived xenografts. Our data suggest that while some cells expressing EMT genes have stem cell activity, not all the EMT states in the mammary gland and breast cancers are associated with an elevated repopulating or tumor initiating capacity.

## Results
### Keratin 14$^+$/S100a4$^+$ cells express genes associated with EMT.
To determine if epithelial cells express EMT associated genes in the adult mammary gland we created a triple transgenic mouse ($Keratin14^{Cre}$/$Rosa26^{TdTomato}$/$S100a4$-GFP) (Fig. 1a), in which we used S100a4, which is expressed by a subset of cells with a

mesenchymal phenotype. S100a4/FSP1 is one of the cell markers associated with EMT that has been linked to increased invasiveness and poor prognoses in breast cancer[20,21]. Moreover, S100a4 knockout mice have significantly reduced metastasis when crossed to mouse models of breast cancer[22] and have been recently used in an attempt to trace EMT cells in breast cancer mouse models[8]. In this mouse model, *Keratin 14-cre*, which marks mammary epithelial cells, drove the expression of Cre in all mammary gland epithelial cells and a Cre-switchable fluorescent marker (*lox-STOP-lox-tdTomato*) is ubiquitously expressed under the control of the β-actin promoter in the *Rosa26* locus. *Keratin 14-Cre* provided increased specificity for mammary epithelial cells and excluded any stromal cells that express high levels of EMT genes from the analysis. S100a4 drove expression of GFP. FACS analysis of the resulting $Keratin14^{Cre}$/$Rosa26^{TdTomato}$/$S100a4$-GFP mice ($n = 6$), showed that lineage$^-$ (CD45$^-$/CD31$^-$/Ter119$^-$/DAPI$^-$) cells consisted of three populations based on the expression of GFP and TdTomato (Fig. 1b, top left panel). 20–60% of lineage$^-$ cells were Keratin-14 TdTomato+/S100a4-GFP$^-$ and corresponded to CD49f$^+$/EpCAM$^+$ luminal and basal cells (Fig. 1b, bottom left panel). Another population was Keratin-14 TdTomato$^-$/S100a4$^-$GFP+ stromal cells, which made up 3–6% of lineage$^-$ cells (Fig. 1b, bottom right panel). Interestingly, there was a small population (0.03–0.1%) of cells that

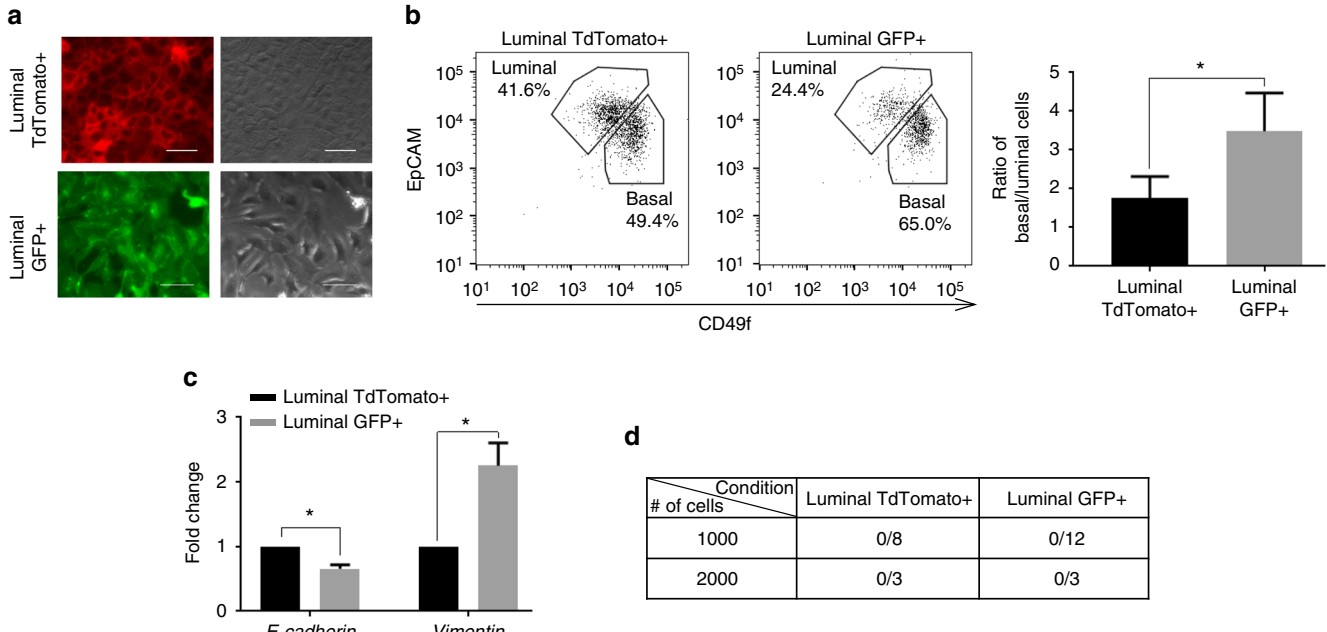

**Fig. 3** Luminal S100a4 traced cells acquire mesenchymal features but cannot transplant in vivo. **a** TdTomato+ and GFP+ luminal cells were sorted from 10–12-week-nulliparous females and plated in 2D culture. TdTomato+ cells show distinct cuboidal morphology (top) while the GFP+ S100a4 lineage traced cells are more mesenchymal like (bottom). Scale bar, 50 μm. **b** FACS analysis of TdTomato+ (left) and GFP+ (middle) luminal cells grown in 3D culture. TdTomato+ luminal cells are plated in 3D culture where some cells switch to GFP+. The organoids are then dissociated and analyzed based on EpCAM/CD49f. Ratio of basal to luminal cells calculated from 3D culture ($n = 3$ cultures derived from three mice, $*p < 0.05$, paired $t$-test). **c** Real time expression analysis of luminal GFP+ compared to TdTomato+ cells isolated from 3D culture. Gene expression values were normalized to *Gapdh*. ($n = 2$ biological replicates). The data are shown as mean ± SD. $*p < 0.05$, $t$-test). **d** Luminal TdTomato+ and GFP+ cells were sorted from 10–12-week-nulliparous females and transplanted in 3-week old mice. Also see Supplementary Fig. 2a. Transplant data are combined from two independent transplant experiments

were double positive (DP) (i.e., Keratin-14 TdTomato+/S100a4-GFP+) consisting of luminal and basal cells (Fig. 1b, top right panel). This data showed that a small population of lineage traced Keratin 14+ epithelial cells are starting to acquire a S100a4-related cellular program.

We next sought to determine if the Keratin 14+/S100a4+ cells expressed EMT genes. As there is no universal marker for EMT and EMT can be described by several states of the cell, we did single-cell gene expression profiling of DP basal cells compared to Keratin 14+/S100a4⁻ basal cells. We found that a higher proportion of the DP cells (21 vs. 3%,) expressed significantly higher levels of *S100a4* (Fig. 1c, Supplementary Fig. 1a, b left most panel, $p = 0.00018$) thus confirming that the S100a4-GFP reporter can enrich in cells expressing the *S100a4* transcript. Moreover, some EMT associated genes, such as *Zeb1* and *Zeb2* were significantly higher by a higher number of the DP cells (Fig. 1c, Supplementary Fig. 1a, b, $p = 0.0002$ and $8.64e^{-08}$, respectively). However, other EMT-associated genes *Sox9*, *Snai1*, and *Snai2*[4] were not expressed significantly higher in the DP cells compared to the TdTomato+/S100a4⁻ cells (Supplementary Fig. 1a, b). In fact our single-cell analysis showed considerable heterogeneity in the expression of genes associated with EMT. For example, we found that either *Snai1*, *Snai2*, or both, were expressed by majority of K14-lineage traced cells in the basal compartment (70% of cells expressed *Snai2*, 14% of cells expressed *Snai1* and 7% expressed both; Supplementary Fig. 1a, b), indicating the role of these genes in maintaining basal cell characteristics[4,11,23,24]. To confirm the single-cell data in a compartment enriched for the ability to transplant the mammary gland, we isolated S100a4⁺ and S100a4⁻ CD49f^high/EpCAM^med (mammary repopulating units; MRUs) and also CD49f^med-low/EpCAM^high (luminal) cells. Gene expression analysis showed that *S100a4*, *Zeb1*, and *Zeb2* are all expressed at significantly higher levels in S100a4+ MRUs and

luminal cells compared to S100a4⁻ cells (Fig. 1d). Thus the DP cells represented a compartment enriched for cells that expressed EMT master regulators *Zeb1* and *Zeb2*. To understand whether expression of EMT-associated genes in this population can enrich for cells with reconstitution potential, we isolated S100a4⁻ MRUs and S100a4+ MRUs and performed in vivo transplantation assays. We found that S100a4+ MRUs have significantly reduced, but still detectable, transplantation potential (Fig. 1e, $p = 0.003$), indicating that expression of *S100a4* and consequently *Zeb1* and *Zeb2* can be uncoupled from transplantation potential in normal mammary homeostasis.

**S100a4 lineage traced basal cells can regenerate the mammary gland in vivo.** The mammary gland develops during puberty by invading the fat-pad to form the mature ductal tree. Hence; we wanted to determine whether S100a4+ cells could contribute to ductal tree development in vivo. Lineage tracing using the Cre system has been used extensively in the mammary gland to study the role of genes in ductal development, if they contribute to mammary epithelial tree and establish lineage hierarchy[25–27]. Hence, we traced the cells using S100a4-Cre, where the S100a4 promoter drives Cre expression in mice embryonically. These mice were crossed to the *Rosa26^mTmG* reporter mice (lox-RFP-STOP-lox-GFP) such that any cell expressing S100a4 and the progenies will be permanently labeled GFP+ (Fig. 2a) embryonically, even if they stop expressing S100a4. Immunofluorescence staining of *S100a4^cre/Rosa26^mTmG* mammary glands revealed that S100a4-GFP+ cells were present both in the luminal and the basal layers of the mammary gland (Fig. 2b). However, the majority of mammary epithelial cells had not switched, indicating that unlike Kertain-14 lineage traced cells, they were not derived from the progeny of the S100a4 traced cells (Fig. 2b,

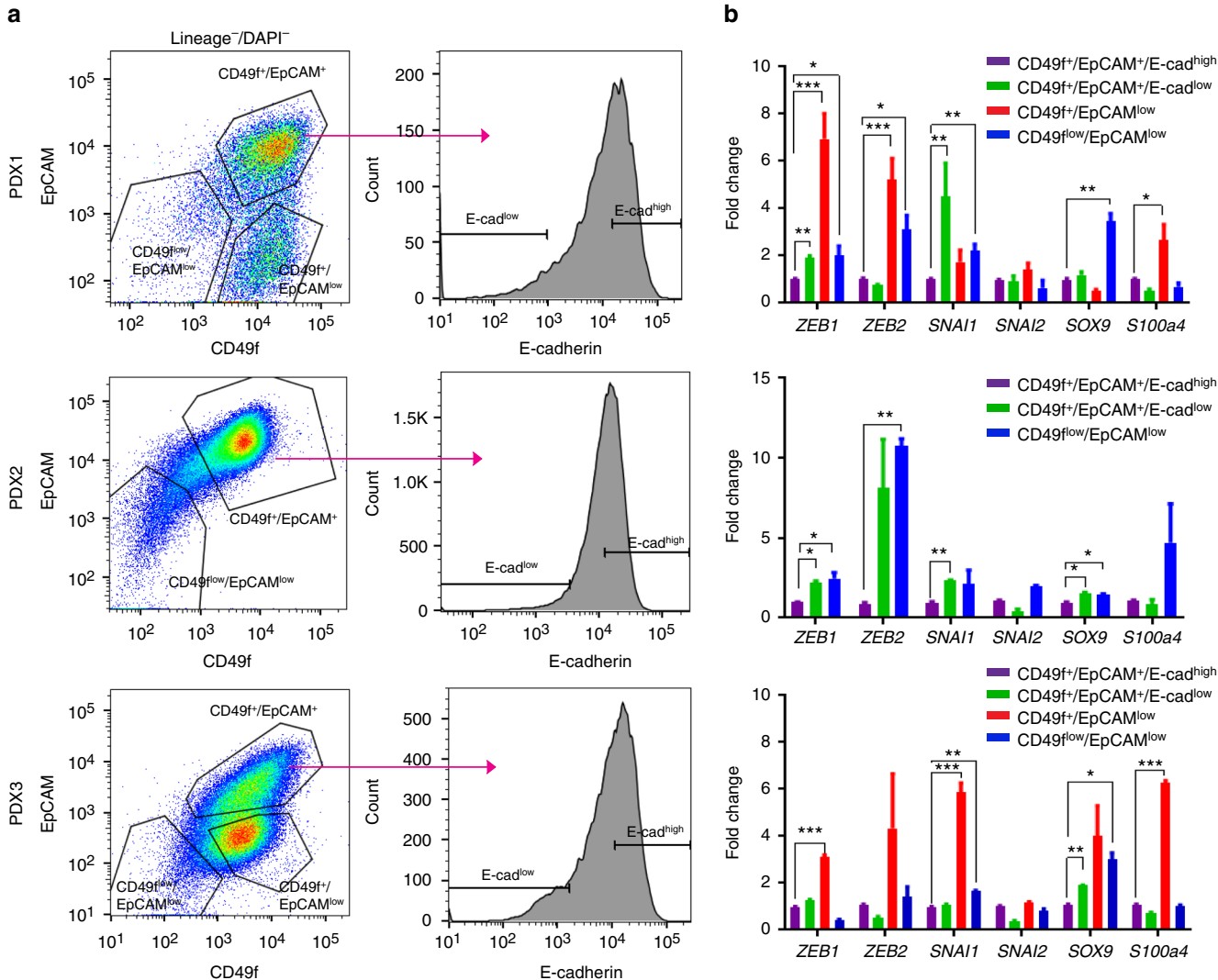

**Fig. 4** Tumorigenic capacity of breast epithelial cells is independent of EMT gene expression. **a** H2kd⁻CD45⁻DAPI⁻ cells from PDX1 (upper), PDX2 (middle), and PDX3 (bottom) stained with CD49f and EpCAM. Sorting gates are depicted in the FACS plot (left panels). The CD49f⁺/EpCAM⁺ population is further divided based on E-cadherin expression (right panels). All populations were double sorted and used in transplantation assays. See Supplementary Fig. 3 for sorting scheme and Table 1. **b** Quantitative gene expression analysis in PDX1 (upper), PDX 2 (middle), and PDX3 (bottom) for CD49f⁺/EpCAM⁺/E-cadherin^high, CD49f⁺/EpCAM⁺/E-cadherin^low, CD49f⁺/EpCAM^low and CD49f^low/EpCAM^low. The data are shown as the mean ± SEM, ($n = 3$ biological replicates in PDX1 and PDX2 and $n = 2$ biological replicates in PDX3). *$p < 0.05$, **$p < 0.01$, ***$p < 0.001$ ($t$-test)

Supplementary Fig. 2a). FACS analysis ($n = 20$) showed that S100a4-GFP+ cells accounted for 0.7–5% lineage⁻ cells (Fig. 2c, Supplementary Fig. 2a). A small percentage of GFP⁺ cells were epithelial (~5%) and consisted of both luminal (CD49f^med-low/Epcam^high) and basal (CD49f^med-high/EpCAM^low-med) lineages (Fig. 2c, d, Supplementary Fig. 2a). We then wanted to compare the transplantation potential of the S100a4 traced cells to the un-traced epithelial cells. To this end, we isolated basal CD49f^med-high/EpCAM^low GFP+ or basal CD49f^med-high/EpCAM^low TdTomato+ cells and performed transplantation assays (Supplementary Fig. 2a). Both GFP+ and TdTomato+ basal cells were able to form mammary outgrowths in vivo that gave rise to luminal and basal cells (Fig. 2e, f). However, the GFP+ cells had lower transplantation potential and formed smaller outgrowths than TdTomato+ cells (Fig. 2e, f, $p = 0.04$). This suggests that some traced cells lost transplantation potential, probably reflecting a more differentiated state. Analysis of TdTomato+ transplants revealed that some transplanted cells turned on S100a4 and became GFP+ in vivo. The switched

GFP+ cells contributed to the mammary tree and gave rise to both luminal and basal lineages (Supplementary Fig. 2b, c). However, the majority of the TdTomato+ cells did not switch but could transplant in vivo. Secondary transplantation revealed that both TdTomato+ and GFP+ cells could be serially passaged (Fig. 2g, h). This showed that some lineage-traced S100a4 cells have the capacity to generate the mammary ductal tree and serially passage in vivo. However, in the basal lineage, cells do not need to express S100a4 to acquire reconstitution ability since both TdTomato+ and GFP+ cells can transplant and have long-term reconstitution potential. The mammary gland develops during puberty by invading the fat pad to form the mature ductal tree. Classical EMT in development is used by cells to migrate during embryonic development. Thus, we predicted that if the mammary epithelial cells used S100a4, Zeb1, and Zeb2 to invade the fat pad, the tree would be traced with S100a4. However, our data suggest that mammary epithelial cells do not need to invoke S100a4/Zeb1/Zeb2 to develop the mammary fat pad during puberty.

**Table 1 Tumorigenicity assay in triple negative patient derived xenografts**

a

| | PDX1 | | | | PDX2 | | PDX3 | | | PDX4 | | PDX5 | |
|---|---|---|---|---|---|---|---|---|---|---|---|---|---|
| | 100 | 500 | 1250 | 3000 | 500 | 1000 | 150 | 300 | 1500 | 500 | 1000 | 100 | 500 |
| CD49f+/EpCAM+/E-cad^high | 4/4 | 4/4 | 4/4 | - | 4/4 | 2/2 | 1/2 | 4/4 | 2/2 | 2/2 | 2/2 | 0/4 | 1/4 |
| CD49f+/EpCAM+/E-cad^low | 0/4 | 0/4 | 2/4 | - | 2/4 | 2/2 | 0/2 | 2/4 | 2/2 | 0/2 | 2/2 | 0/4 | 1/4 |
| CD49f+/EpCAM^low | 0/4 | 0/4 | 0/4 | 1/2 | - | - | - | 1/4 | 2/2 | 0/2 | 2/2 | - | - |
| CD49f^low/EpCAM^low | 0/4 | 0/4 | 0/4 | - | 0/4 | 0/4 | 0/2 | 1/4 | 1/2 | 0/2 | 0/2 | 0/4 | 0/4 |

b

| | PDX1 | PDX2 | PDX3 | PDX4 | PDX5 |
|---|---|---|---|---|---|
| | Frequency | Frequency | Frequency | Frequency | Frequency |
| CD49f+/EpCAM+/E-cad^high | 1 in 71 (1/31–1/131) | 1 in 290 (1/176–1/478) | 1 in 115 (1/67–1/197) | 1 in 372 (1/208–1/666) | 1 in 2140 (1/972–1/5883) |
| CD49f+/EpCAM+/E-cad^low | 1 in 3,032 (1/1513–1/6077) | 1 in 532 (1/312–1/905) | 1 in 504 (1/289–1/882) | 1 in 910 (1/451–1/1836) | 1 in 2140 (1/972–1/5883) |
| CD49f+/EpCAM^low | 1 in 11,837 (1/4450–1/31853) | - | 1 in 675 (1/363–1/1253) | 1 in 910 (1/451–1/1836) | - |
| CD49f^low/EpCAM^low | Did not calculate as no tumors were formed | | 1 in 1175 (1/841–1/3618) | Did not calculate as no tumors were formed | |

(a) H2kd−CD45−DAPI− cells from five different PDX were double sorted for different population as indicated (also see Fig. 4) and the indicated number of cells were injected in mice. Mice were checked every week to determine if tumors were present. No mice were excluded from analysis. Mice were euthanized when tumors reached 1.5 cm in the longest dimension. No predetermined experimental endpoint based on time was set. Bold numbers indicate tumor sub-populations that had at least 1 positive outgrowth
(b) L-calc was used to calculate tumor initiating frequency within each sorted population for every tumor. *$p < 0.05$, ****$p < 0.0001$

**Luminal S100a4 traced cells acquire mesenchymal features but cannot transplant in vivo.** The reconstitution ability of the mammary gland is largely confined to the basal epithelium[28–30]. However, S100a4+ cells are present in both basal and luminal layers. In previous literature, the acquisition of mesenchymal features in epithelial cells has been linked to increased tumor initiating capacity and repopulating ability[3–5,11]. Previous studies have shown that expression of S100a4 leads to acquisition of some mesenchymal characteristics in the mammary carcinoma cells[8,22]. Consistent with these studies, we found that S100a4 traced luminal cells changed their morphology from cobblestone epithelial to spindle shaped mesenchymal-like in 2D cultures (Fig. 3a). We also cultured luminal cells in 3D matrigel assays that more closely resembled in vivo conditions[31]. FACS analysis of 3D colonies using CD49f and EpCAM showed that compared to the luminal TdTomato+ cells; S100a4 traced luminal cells have more basal characteristics (higher basal/luminal ratio) and lower expression of EpCAM (Fig. 3b, $p = 0.02$). Gene expression profiling of TdTomato+ and switched GFP+ cells from the 3D colonies revealed that S100a4 traced cells express lower levels of *E-cadherin* and higher levels of *Vimentin* (Fig. 3c). Hence we investigated whether S100a4 traced luminal cells gain reconstitution ability in vivo. We sorted GFP+ CD49f^low-med/EpCAM^high and TdTomato+ CD49f^low-med/EpCAM^high luminal cells from 10–12 week mice and performed transplantation assays

(Supplementary Fig. 2a). We did not observe any outgrowths from the TdTomato+ or GFP+ CD49f^low-med/EpCAM^high luminal cells (Fig. 3d). These data demonstrated that luminal S100a4 traced cells acquired mesenchymal characteristics but did not gain reconstitution ability. This suggested that the expression of *S100a4* and consequently, *Zeb1* and *Zeb2* in the luminal cells is largely a stochastic event and not sufficient for cells to gain reconstitution capacity in transplantation assays. These data were also consistent with the observation that enforced expression of *Sox9* and *Snai2* does not induce reprogramming of most luminal cells[4].

**Tumorigenic capacity of breast epithelial cells is independent of EMT gene expression.** We and others have previously demonstrated that tumorigenic capacity in cancer is enriched in a subset of cells within the tumor[32,33]. It has been suggested that breast tumor cells that undergo EMT in vitro acquire tumor initiating capacity, and the expression of EMT genes in breast cancer cells increases tumor initiating frequency[10,11]. Recent studies have also shown considerable heterogeneity within tumor cells, where tumor cells express mesenchymal genes and can transition between epithelial and mesenchymal states[16]. As we have previously demonstrated that patient derived xenografts (PDX) faithfully recapitulate major aspects of primary human

tumors[18], we used ER⁻/PR⁻/Her2⁻ (Triple negative breast cancer; TNBC) PDX models to test if cells expressing EMT associated genes had higher tumor initiating capacity. We used EpCAM and CD49f that have been previously utilized to separate both normal human mammary epithelial cell and also breast cancer cells into different populations[34,35]. We isolated 3 different populations; CD49f$^{low}$/EpCAM$^{low}$, CD49f⁺/EpCAM$^{low}$ and CD49f⁺/EpCAM⁺ (Fig. 4a, left panel, Supplementary Fig. 3). The CD49f⁺/EpCAM⁺ population was further subdivided into CD49f⁺/EpCAM⁺/E-cadherin$^{low}$ and CD49f⁺/EpCAM⁺/E-cadherin$^{high}$ (Fig. 4a, right panel) and used for tumorigenicity assays. We found that in most PDXs tumor initiating capacity was enriched in the CD49f⁺/EpCAM⁺ population (Table 1a, b). Within this population in four of the five tumors, E-cadherin$^{high}$ cells had higher tumor initiating capacity compared to E-cadherin$^{low}$ cells (Table 1a, b). Real time gene expression analysis of the different populations showed that CD49f⁺/EpCAM⁺/E-cadherin$^{low}$, CD49f⁺/EpCAM$^{low}$, and CD49f$^{low}$/EpCAM$^{low}$ cells expressed higher levels of the EMT associated genes (ZEB1, ZEB2, and SNAI1) (Fig. 4b). However, the higher expression of these genes did not translate into enriched tumorigenic capacity in transplantation assays (Table 1a, b). Furthermore, analysis of the tumors from the different population showed that in most cases CD49f⁺/EpCAM⁺/E-cadherin$^{low}$ and CD49f⁺/EpCAM⁺/E-cadherin$^{high}$ cells were able to give rise to all populations in the tumor (Supplementary Fig. 4) suggesting some plasticity between the two populations. However, in some tumors such as PDX1 and PDX3 we found that CD49f⁺/EpCAM$^{low}$ and CD49f$^{low}$/EpCAM$^{low}$ cells gave rise to tumors that had a CD49f⁺/EpCAM$^{low}$ profile, indicating that at least some cells in this population have less plasticity (Supplementary Fig. 4, column 3 and 4). However, these cells did not form characteristic mesenchymal tumors (Supplementary Fig. 5). Furthermore, in contrast to normal mammary epithelial cells were the repopulating ability is enriched in the CD49f⁺/EpCAM$^{med-low}$ population, we find that in TNBC samples tumor initiating capacity is highest in CD49f⁺/EpCAM⁺ population. TNBC tends a higher mutation burden compared to estrogen receptor positive tumors[36–38]. The higher rate of mutations could perturb the marker expression of the tumorigenic cells, or could confer self-renewal properties to cell populations that don't do so in the normal duct[39]. Single-cell gene expression profiling of human primary normal mammary epithelial cells and TNBC samples showed that the CD49f⁺/EpCAM⁺ cells in tumors neither resemble the luminal progenitors nor the CD49f⁺/EpCAM$^{med-low}$ stem cells of the normal mammary gland (Supplementary Fig. 6). This suggests that in many cases mutations in triple negative breast cancer give rise to cells whose phenotype materially differs from normal mammary gland cells.

## Discussion

In conclusion, our data suggest that higher endogenous levels of some EMT associated genes are not sufficient to confer increased repopulating ability in the normal mammary gland. Recent studies have suggested that EMT is a series of cell states transitioning between epithelial to mesenchymal phenotypes. Using single-cell analysis we find that there is considerable heterogeneity in cells expressing EMT associated genes in the normal mammary gland. Specifically, ~70% of basal cells express Snai2, ~14% express Snai1 and only a minority of cells express Zeb1 and Zeb2. Previous studies have suggested that normal mammary basal cells do not express Snai1 and Zeb1[11]. Using S100a4 as a surrogate, our model can isolate rare Zeb1+ cells and study their contribution in vivo by lineage tracing and transplantation. Our model can then test the transplantation potential of cells that express Snai2/Snai1 and those that express Snai1/Snai2/Zeb1/Zeb2. We find

that some cells within the S100a4+/Zeb1+/Zeb2+ population have limited reconstitution ability. Although, our model can significantly enrich rare Zeb1+/Zeb2+ cells in the normal mammary gland, it is possible due to limitations of using a single marker such as S100a4 and inefficiencies of the Cre reporter that we may not trace every cell expressing Zeb1/Zeb2[40,41]. It is important to note that overall, basal cells express keratins and smooth muscle actin but have lower levels of E-cadherin compared to luminal cells. Therefore, it can be said that the reconstitution potential of basal cells is inherent in their 'quasi-mesenchymal' state[24]. Moreover, we have recently demonstrated that quiescent Bcl11b+ cells[42] in the basal layer do not express EMT associated genes such as Zeb1/Zeb2 but are required for mammary gland regeneration and development. This further confirms our data that repopulating ability in normal mammary development does not require expression of EMT associated genes, S100a4/Zeb1/Zeb2. Also, we find that in contrast to recent findings[43] S100a4+/Zeb1+/Zeb2+ cells have less plasticity in mammary epithelial cells, suggesting tissue specificity of different genes. This is also consistent with our PDX data where increasing expression of ZEB1/ZEB2/SNAI1 leads to lower tumor initiating capacity suggesting that a terminal mesenchymal phenotype has the lowest tumor initiating frequency. Our data also suggest that although PDX's express ZEB1, ZEB2, and SNAI1, the expression of these genes alone does not confer tumorigenicity to the cells. On the contrary, in 4/5 TNBC PDXs that we tested, CD49f⁺/EpCAM⁺/E-cadherin$^{high}$ cells have the highest tumor initiating capacity. This suggests that at least in some TNBCs, tumorigenic capacity is not dependent on the level of expression of EMT associated genes. Our result reveals that there are two populations of tumor-initiating cells: one population that has a higher expression of EMT associated genes and one population that has a lower expression of EMT associated genes. In addition, our data imply that both the E-cadherin$^{high}$ and E-cadherin$^{low}$ populations can regenerate the other population. It is possible that the E-cadherin$^{low}$ cells may represent a transient state of partial EMT[44] and the switching between the two states may be important for metastatic dissemination and metastatic colonization[9]. However, CD49f⁺/EpCAM$^{low}$ that express highest levels of ZEB1/ZEB2/SNAI1 have the fewest number of cells with the necessary plasticity to transition to other cell populations. In summary, expression of EMT associated genes does not enrich for cells capable of reconstitution in both normal mammary epithelial cells and in PDX models, but may have roles in other processes. This has important implications for designing therapy and in the future, it would be important to study the role of different populations in tumor maintenance and metastasis.

## Methods

**Mice**. S100a4-Cre transgenic mice (Stock #012641), K14-Cre transgenic mice (Stock #004782), S100a4-GFP (Stock #012904), Rosa26$^{mTmG}$ (Stock #007676), Rosa26$^{TdTomato}$ (Stock #007914) and Nod/Scid/Il2r−/− (NSG) (Stock #005557) mice were purchased from the Jackson Laboratory. The resulting mice from crosses were on mixed backgrounds and hence for recipient mice in the transplant assays we used NSG mice. All mice used for this study were maintained at Stanford Animal Facility in accordance with the guidelines of the animal care use committee (APLAC #10868).

**Preparation of single-cell suspensions of mammary gland**. 10–12-week-old virgin female mice were killed and fat pads 2, 3, and 4 were surgically resected. Tissue was digested in DMEM/F12 for 2 h, and then processed as previously described[45]. Briefly, mechanically dissociated mammary glands were treated 2 h with collagenase and hyaluronidase (StemCell Technologies, Inc.) followed by lysis of red blood cells in ACK (NH4CL) for 5 min, then 1–2 min treatment with pre-warmed 0.25% trypsin EDTA and finally treatment with prewarmed dispase (StemCell Technologies, Inc) plus DnaseI (Sigma) for 2 min. The cells were then filtered through a 40 µm mesh and washed with flow cytometry buffer (HBBS, 2% FBS, PSA).

**Patient derived xenografts**. For human samples, informed consent was obtained after the approval of protocols by the Stanford University and City of Hope Institutional Review Boards (IRB #4344). Human breast specimens, primary or xenograft tumors were mechanically dissociated into >1–2 mm³ pieces with a razor blade and digested at 37 °C with collagenase and hyaluronidase, in Advanced DMEM/F12 (Invitrogen) with 2 mM Glutamax (Invitrogen), 120 µg/mL penicillin, 100 µg/mL streptomycin, 0.25 µg/mL amphotericin-B (PSA) and allowed to incubated for 3–4 h with pipetting. At the end of the incubation cells were treated with ACK to lyse the red blood cells followed by a short incubation of Dispase and DNaseI. (Note: No trypsin is used in processing human PDX samples). The cells were filtered through a 70 µm nylon mesh and washed with flow cytometry buffer (HBBS, 2% FBS, PSA).

**Flow cytometry**. To reduce non-specific binding, cells suspended in staining buffer were blocked on ice for 10 min with rat IgG (Sigma) 10 mg/mL at 1:1000. The cells were then stained, in the dark, on ice for 30 min with optimal antibodies concentrations, which was determined by titration experiments. List of antibodies, clones and dilutions are provided in Supplementary Table 3. Lineage cells consist of CD45, Ter119, and CD31 positive cells in mouse. In PDX lineage cells consist of cell positive for CD45 and H2Kd. Flow Cytometry was performed with 130 µm nozzle on a BD Flow cytometry Aria II with Flow cytometry Diva software. The data analysis was performed using Flowjo. For all experiments, side scatter and forward scatter profiles (area and width) were used to eliminate debris and cell doublets (Supplementary Fig. 3). Dead cells were eliminated by excluding 4′,6-diamidino-2-phenylindole (DAPI) positive cells (Molecular Probes).

**In vivo transplants**. Sorted live lineage^neg TdTomato⁺ or GFP⁺ luminal and basal cells were collected in staining media and resuspended with 33% matrigel. Per transplant 10 µL was injected into cleared fat pads of weaning age C57Bl/6 mice (21–28 days) as previously described[46]. All transplants were allowed to grow for 8 weeks before analysis.

**Xenograft tumor cell infection and engraftment**. For tumorigenecity assays xenograft cells were double sorted, purity checked and resuspended in staining buffer and injected with 50% matrigel in fourth abdominal fat pad by subcutaneous injection at the base of the nipple of female NSG mice. The mice were monitored every week for tumor growth.

**In vitro colony formation assay**. L1-wnt3A feeder cells (generous gift from Roel Nusse lab) were administered a 40 Gy dose of X-ray irradiation and mixed with growth factor reduced matrigel at a density of 10 k/70 µL matrigel/well. 1000 TdTomato⁺ luminal cells per well were resuspended in 200 µL culture media (DMEM F12 + 2% FBS + PSA + B27 + 10 mM HEPES) supplemented with EGF (10 ng/mL, BD Bioscience), Rspo1(250 ng/mL, R&D), ROCK inhibitor Y27632 (10 µM, Sigma), and Noggin (100 ng/mL, R&D), and were overlaid on top of the matrigel. Plate was maintained in 37° incubator at 5% CO₂ for 1–2 weeks. The colonies were recovered using trypsin and sorted based on DAPI, GFP, TdTomato, Epcam, and CD49f.

**Real time PCR**. 500–2000 primary mammary cells of various populations were directly sorted into Eppendorf tube containing 400 µL Trizol LS (Life Technologies). RNA was extracted according to the manufacturer's instruction with addition of ultrapure glycogen (Life Technologies) as carrier. RNA was reverse transcribed to cDNA using SuperScript III First Strand Synthesis kit (Life Technologies) according to the manufacturer's instructions. cDNA was preamplified 15–20 cycles according to the cell number using TaqMan pre-amp mastermix (Applied Biosystems) and target gene Taqman primer pool. Preamplified cDNA was then subjected to the real time PCR for specific gene target according to manufacturer's instruction using 7900HT Real Time PCR system (Applied Biosystems). List of Taqman probes is shown in Supplementary Table 2. All expression data are normalized to *Actnb* and *Gapdh*. The data were analyzed by SDS2.4 software and Excel.

**Immunofluorescence**. Mammary glands were fixed in formalin and embedded in paraffin for immunostaining. Sections were de-paraffinized, dehydratated, and microwaved for 10 min at 95 °C in Tris–EDTA (pH 9) for antigen retrieval. The tissue sections were incubated o/n at 4 °C with primary antibodies diluted in PBS + 5% BSA. The samples were incubated with anti-GFP Alexa-488 (1:500), anti-Keratin-14 (1:100), anti-Keratin-8 (1:100), anti-rabbit and anti-rat Alexa-594 conjugated secondary antibodies (Invitrogen) at 1:500 in PBS + 5% BSA 1 h at RT. All the immunofluorescence sections and cells were mounted in ProLong Gold with DAPI. Images were acquired by Carl Zeiss LSM 510 Meta confocal microscope. Images were processed using ImageJ.

**Single-cell gene expression analysis**. This was performed as described before[18]. In short double sorted single cells were sorted into individual wells of 96 plates containing 5 µL lysis buffer (cellsDirect qRT-PCR mix; Invitrogen) and 2U SuperaseIn. After reverse transcription and pre-amplification using multiplexed

PCR, reactions were loaded (Hamilton StarLET pipetting robot) on a fluidigm microfluidic chip. List of Taqman probes is shown in Supplementary Table 2. The microfluidic chips were run on the BioMark real-time PCR reader (Fluidigm), and loaded chips underwent thermocycling and fluorescent quantification according to manufacturer instructions. This resulted in 48 or 96 gene-expression values (measured in threshold cycles, Ct) for each one of the cells sorted. For clustering analysis, we standardized the expression levels of each gene individually by subtracting the mean and dividing by three times the standard deviation of expressing cells. Then, all values were truncated into the range [−1, +1] as previously described (Dalerba et al.[18]). Clustering was performed using complete linkage and correlation distance (Matlab). Positive or negative associations among pairs of genes were tested by Spearman correlation, and *p*-values were calculated using the Wilcoxon Rank-sum test.

**Statistical analysis**. All graphs show the average as central values and error bars indicate ± SD unless otherwise indicated. *P*-values are calculated using the *t*-test, Wilcoxon Rank-sum test and paired *t*-test as indicated in the figure legends. All *p*-values are calculated using graphpad prism software. For calculating frequency and *p*-values in mammary transplantion assays and tumorigenicity studies, ELDA and L-calc software from Stem Cell technologies are used respectively. For single-cell gene expression analysis the Wilcoxon Rank-sum test was used to calculate *p*-values. For animal studies, sample size was not predetermined to ensure adequate power to detect a pre-specified effect size, no animals were excluded from analyses, experiments were not randomized and investigators were not blinded to group allocation during experiments.

**Data availability**. The authors declare that all other relevant data supporting the findings of the study are available in the article and its supplementary information files, or from the corresponding author upon request.

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

## Acknowledgements

This work was supported by NIH/NCI grant 5P01 CA139490-05, NIH/NCI grant 5U01 CA154209-04, NIH/NCI 5R01 CA100225-09, Department of Defense grant W81XWH-11-1-0287, Department of Defense/Breast Cancer Research Program (BCRP) Innovator Award W81XWH-13-1-0281, DoD Post-doctoral Fellowship W81XWH-12-1-0020 to S.S. We thank Patty Lovelace for her assistance with flow cytometry and Stanford Neuroscience Microscopy Service, supported by NIH NS069375. BD FACSAriaII was purchased by NIH S10 shared instrumentation grant 1S10RR02933801.

## Author contributions

S.S.S. designed, performed and analyzed research and wrote the paper; T.K., analyzed research; S.S.S., A.H.K., S.C., N.A.L., R.W.H., M.Z., D.Q., and F.A.S. derived and characterized the patient derived xenografts; S.S. performed the single-cell gene expression profiling. F.M.D., G.S., helped in obtaining patient samples and S.R.Q helped with single cell analysis. M.F.C. designed the research and wrote the paper.

## Additional information

**Competing interests:** The authors declare no competing financial interests.

