## [Peer Review File · Nature Communications]

Reviewers' comments:

Reviewer #2 (Remarks to the Author):

The authors have satisfactorily addressed the concerns I raised when this manuscript was previously submitted . However I would like to see them add the data included in Fig 1 in the rebuttal letter which compares single cell gene expression in PDX and normal mammary cells. In addition, the authors still to provide additional evidence for the use of S100a4 by itself as a definitive marker of EMT.

Reviewer #3 (Remarks to the Author):

While I believe that the experiments are well performed and the data are valuable, I think that the concepts are not novel but valuable to be published in Nat Comm., as they confirm that EMT and stemness are not always linked and that epithelial /epitheloid cells have higher metastatic potential. This is a very important concept and this work shows it in a very elegant manner. However, the way the results are discussed is confusing and can be misleading. As mentioned, it is now clear that EMT and stemness are not always associated and indeed, that epitheloid cells have higher metastatic potential than full mesenchymal cells. This is in agreement with the data presented. However, there are alternative interpretations that fit with these and other published data. For instance, Beerling et al. Cell Rep 2016, although refereed acn also be discussed as how a pool of breast tumor cells undergo EMT and disseminate to then reverse to the epithelial state for metastatic outgrowth. This is also compatible with the requirement of Twist downregulation for metastatic colonization (Tsai et al, Cancer Cell 2012). Furthermore, other transcription factors that can induce EMT need to be downregulated for metastatic outgrowth (e.g. Snail1, see Tran et al, Cancer Res 2014). Also compatible with the work by Beck et al (Cell Stem Cell 2015), which shows that low levels of Twist are associated with tumor initiating capacities, while higher levels are not, but are required for the induction of EMT (progression as discussed by the authors). Thus, levels matter, and low levels (that can be obtained after downregulation of higher levels required for dissemination) are associated with tumour initiating (cancer stem cell) properties, and cell plasticity lies at the core of this dynamic process. As the authors also mention the work by Fisher et al Nature 2015 and also Krebs et al Nat Cell Biol, 2017, in terms of EMT and metastasis, they may want to consider some of the concepts discussed in Ye et al .and Aiello et al both in Nature 2017 and in the N&V published with Krebs et al., 2017.

Reviewer #4 (Remarks to the Author):

The major issue with this manuscript is the assumption that EMT is being reported by the expression of S100a4. This assumption is poorly supported by the data in the manuscript. This issue is raised by 2 previous reviewers but has not been dealt with by the authors in a full and frank way. They appear to be in denial of the way thinking in the field has progressed. The discussion should include, at the very least, a frank admission that S100a4 may not be a faithful

reporter of EMT and that all papers making this assumption are potentially flawed. In fact the nebulous nature of EMT that emerges from the revised version adds a straw-man aspect to the manuscript's underlying hypothesis- maybe EMT never was an issue in stemness and showing that EMT is irrelevant really doesn't mean much.

Reviewers' comments: Reviewer #2 (Remarks to the Author):

The authors have satisfactorily addressed the concerns I raised when this manuscript was previously submitted. However I would like to see them add the data included in Fig 1 in the rebuttal letter which compares single cell gene expression in PDX and normal mammary cells. In addition, the authors still to provide additional evidence for the use of S100a4 by itself as a definitive marker of EMT.

As requested by the Reviewer we have added Figure 1 from the rebuttal letter to the main manuscript (Supplementary Figure 6).

As for the second question: Throughout our manuscript we refer to the S100a4+ cells as cells expressing “EMT associated genes” and not cells that have undergone EMT. Moreover, we have modified the text to more specifically highlight the limitations of using S100a4. S100a4 expression enriches for cells expressing Zeb1 and Zeb2 (Figure 1 and Supplementary Figure 1). Also, lineage traced S100a4 luminal cells acquire mesenchymal features *in vitro* (Figure 3). We understand that S100a4 is probably not representative of every cell expressing EMT genes in the mammary gland as seen by the heterogeneity seen in expression of EMT genes by our own single cell data. After the two Nature papers^{1,2} suggesting that EMT is not required for metastasis there is a debate whether S100a4, Vim or SMA are EMT markers^{3,4}. However, there is a general consensus that cells undergoing EMT express lower levels of E-cadherin. Hence, we tested whether S100a4+ cells express lower levels of E-cadherin. Indeed we find that both S100a4+ cells and lineage traced cells express lower levels of E-cadherin (Figure 1). Importantly, this is consistent with our human xenograft data where the CD49f⁺/EpCAM^{low} cells express higher levels of S100a4 (Figure 4 in the manuscript).

Figure 1: (a) FACS analysis of S100a4-GFP mice. Lineage-/live/Epithelial cells (left most panel) were then divided based on expression of GFP (second from the left) and measured for surface expression of E-cadherin (right most panels) n=2 mice. (b) Real-time PCR analysis of S100a4+ epithelial cells and S100a4- epithelial cells for *Cdh1*. All values are normalized to *Actin* and *Gapdh* n= 2 mice (c) Real-time PCR analysis of mTmG^{S100a4-cre} mice where traced GFP+ epithelial cells are compared to TdTomato+ epithelial cells. n= 4 mice. All values are normalized to *Actin* and *Gapdh*. (Note: The difference in *Cdh1* by real-time PCR is less in the lineage traced cells reflecting the ability of some of them to become epithelial again).

Reviewer #3 (Remarks to the Author):

While I believe that the experiments are well performed and the data are valuable, I think that the concepts are not novel but valuable to be published in Nat Comm., as they confirm that EMT and stemness are not always linked and that epithelial /epitheloid cells have higher metastatic potential. This is a very important concept and this work shows it in a very elegant manner. However, the way the results are discussed is confusing and can be misleading. As mentioned, it is now clear that EMT and stemness are not always associated and indeed, that epitheloid cells have higher metastatic potential than full mesenchymal cells. This is in agreement with the data presented. However, there are alternative interpretations that fit with these and other published data. For instance, Beerling et al. Cell Rep 2016, although refereed can also be discussed as how a pool of breast tumor cells undergo EMT and disseminate to then reverse to the epithelial state for metastatic outgrowth. This is also compatible with the requirement of Twist downregulation for metastatic colonization (Tsai et al, Cancer Cell 2012). Furthermore, other transcription factors that can induce EMT need to be downregulated for metastatic outgrowth (e.g. Snail1, see Tran et al, Cancer Res 2014). Also compatible with the work by Beck et al (Cell Stem Cell 2015), which shows that low levels of Twist are associated with tumor initiating capacities, while higher levels are not, but are required for the induction of EMT (progression as discussed by the authors). Thus, levels matter, and low levels (that can be obtained after downregulation of higher levels required for dissemination) are associated with tumour initiating (cancer stem cell) properties, and cell plasticity lies at the core of this dynamic process.

As the authors also mention the work by Fisher et al Nature 2015 and also Krebs et al Nat Cell Biol, 2017, in terms of EMT and metastasis, they may want to consider some of the concepts discussed in Ye et al .and Aiello et al both in Nature 2017 and in the N&V published with Krebs et al., 2017.

We thank the reviewer for their kind words about our work and apologize for anything that is misleading in the paper. We hope that with this revision we have corrected that. While we agree that downregulation of epithelial characteristics is very likely to be a requirement for metastatic dissemination. However, recent studies have synonymized EMT to stem cells in breast cancer. This is a problem in studying cancer stem cells and self-renewal properties of these cells. Our xenograft data shows that the most mesenchymal cell populations CD49f⁺/EpCAM^{low} have low tumorigenicity and do not retain the plasticity seen in the CD49f⁺/EpCAM⁺ cells. The cells that have this dynamic plasticity are in the most epithelial compartment of the tumor. It is possible that these cancer stem cells downregulate epithelial characteristics during dissemination but that is not the same as acquiring stem cell characteristics or properties of self-renewal. Therefore, in our opinion while there can be some cells within those in the EMT states that have stem cell activity; EMT is not a requirement to acquire stem cell properties in cancer. This is also demonstrated well in the normal mammary gland where deletion of Snai1⁵, Snai2⁶ or Twist1⁷ doesn't not impact the development of the mammary tree. Finally, as suggested by the reviewer we have discussed the concepts suggested in the discussion of the paper.

Reviewer #4 (Remarks to the Author):

The major issue with this manuscript is the assumption that EMT is being reported by the expression of S100a4. This assumption is poorly supported by the data in the manuscript. This issue is raised by 2 previous reviewers but has not been dealt with by the authors in a full and frank way. They appear to be in denial of the way thinking in the field has progressed. The discussion should include, at the very least, a frank admission that S100a4 may not be a faithful reporter of EMT and that all papers making this assumption are potentially flawed. In fact the nebulous nature of EMT that emerges from the revised version adds a straw-man aspect to the manuscript's underlying hypothesis- maybe EMT never was an issue in stemness and showing that EMT is irrelevant really doesn't mean much.

The authors thank the reviewer for the comments and apologize for any confusion in the text of the previous submission regarding our data. We fully agree that a single marker such as S100a4 or for that matter any gene, is not representative of an entire biological process, in this case EMT. Our single cell data gene expression data clearly shows this to be true. We show using single cell analysis that S100a4+ cells are enriched in expression of Zeb1 and Zeb2 (Figure 1 and Supplementary Fig 1), which have been previously implicated in EMT. This is also true in patient samples where in expression of S100a4 increases with decreasing levels of EpCAM and E-cadherin (Figure 4). Our single cell data also shows that there is great heterogeneity in cells that express EMT associated genes, showing that there is not a single mesenchymal cell state. Hence in the manuscript (including the title) we use the term '*EMT-associated genes*' and in most cases refer to the gene itself. Moreover, as suggested by the reviewer we have modified the discussion to include the limitations of our system and have corrected all places in the text that indicate that S100a4 is sole marker for EMT. We are sorry for not being clearer in the original version.

We also agree that EMT does not necessarily lead to the stem cell state and targeting EMT in cancer stem cells likely will not by itself lead to curative treatments. However, a quick literature search will demonstrate reports with the opposite conclusion and we have cited a number of these papers in our manuscript.

The major focus of the laboratory is understanding the molecular regulation of stem cells. We do think our data clearly shows that some, but not all, cells with mesenchymal characteristics can behave as stem cells. Furthermore, some, but not all, cells with an epithelial phenotype behave as stem cells. Moreover, each subpopulation of mesenchymal cells and epithelial cells that read out as stem in our assays, both lineage tracing and transplantation, can give rise to the other. It is of course always possible that some unknown factor(s) could induce plasticity in other cell populations. That hypothesis is impossible to disprove.

REFERENCES

- 1 Fischer, K. R. *et al.* Epithelial-to-mesenchymal transition is not required for lung metastasis but contributes to chemoresistance. *Nature* **527**, doi:10.1038/nature15748 (2015).
- 2 Zheng, X. *et al.* Epithelial-to-mesenchymal transition is dispensable for metastasis but induces chemoresistance in pancreatic cancer. *Nature* **527**, doi:10.1038/nature16064 (2015).
- 3 Ye, X. *et al.* Upholding a role for EMT in breast cancer metastasis. *Nature* **547**, E1-E3, doi:10.1038/nature22816 (2017).
- 4 Aiello, N. M. *et al.* Upholding a role for EMT in pancreatic cancer metastasis. *Nature* **547**, E7-E8, doi:10.1038/nature22963 (2017).
- 5 Ni, T. *et al.* Snail1-dependent p53 repression regulates expansion and activity of tumour-initiating cells in breast cancer. *Nat Cell Biol* **18**, 1221-1232, doi:10.1038/ncb3425<http://www.nature.com/ncb/journal/v18/n11/abs/ncb3425.html#supplementary-information> (2016).
- 6 Nassour, M. *et al.* Slug Controls Stem/Progenitor Cell Growth Dynamics during Mammary Gland Morphogenesis. *PLoS ONE* **7**, e53498, doi:10.1371/journal.pone.0053498 (2012).
- 7 Xu, Y. *et al.* Inducible Knockout of *Twist1* in Young and Adult Mice Prolongs Hair Growth Cycle and Has Mild Effects on General Health, Supporting *Twist1* as a Preferential Cancer Target. *The American Journal of Pathology* **183**, 1281-1292, doi:10.1016/j.ajpath.2013.06.021.

REVIEWERS' COMMENTS:

Reviewer #2 (Remarks to the Author):

The authors have satisfactorily addressed my concerns

Reviewer #4 (Remarks to the Author):

The authors now include the followings statement in the Discussion and refer to the "EMT associated genes".

Although, our model can significantly enrich rare Zeb1+/Zeb2+ cells in the normal mammary gland, it is possible due to limitations of using a single marker such as S100a4 and inefficiencies of the Cre reporter that we may not trace every cell expressing Zeb1/Zeb2^{42,43}.

This is recognition of the limits of the tracing technique and the emphasis on diversity within the traced cell population serves to reinforce the message. More robust critical discussion of the technical limitations of single gene tracing, presented here and in the published works of others, would be appropriate, but what is provided is sufficient to convey the message.